# COMBSecretomics: A pragmatic methodological framework for higher-order drug combination analysis using secretomics

**Efthymia Chantzi**[1,2]*, **Michael Neidlin**[3], **George A. Macheras**[4], **Leonidas G. Alexopoulos**[3], **Mats G. Gustafsson**[1,2]*

**1** Cancer Pharmacology and Computational Medicine, Department of Medical Sciences, Uppsala University, Uppsala, Sweden, **2** Signals and Systems, Department of Electrical Engineering, Uppsala University, Uppsala, Sweden, **3** Biomedical Systems Laboratory, Department of Mechanical Engineering, National Technical University of Athens, Athens, Greece, **4** 4th Orthopedic Department, General Hospital KAT, Athens, Greece

* efthymia.chantzi@medsci.uu.se (EC); mats.gustafsson@medsci.uu.se (MGG)

## Abstract

Multi drug treatments are increasingly used in the clinic to combat complex and co-occurring diseases. However, most drug combination discovery efforts today are mainly focused on anticancer therapy and rarely examine the potential of using more than two drugs simultaneously. Moreover, there is currently no reported methodology for performing second- and higher-order drug combination analysis of secretomic patterns, meaning protein concentration profiles released by the cells. Here, we introduce COMBSecretomics (https://github.com/EffieChantzi/COMBSecretomics.git), the first pragmatic methodological framework designed to search exhaustively for second- and higher-order mixtures of candidate treatments that can modify, or even reverse malfunctioning secretomic patterns of human cells. This framework comes with two novel model-free combination analysis methods; a tailor-made generalization of the highest single agent principle and a data mining approach based on top-down hierarchical clustering. Quality control procedures to eliminate outliers and non-parametric statistics to quantify uncertainty in the results obtained are also included. COMBSecretomics is based on a standardized reproducible format and could be employed with any experimental platform that provides the required protein release data. Its practical use and functionality are demonstrated by means of a proof-of-principle pharmacological study related to cartilage degradation. COMBSecretomics is the first methodological framework reported to enable secretome-related second- and higher-order drug combination analysis. It could be used in drug discovery and development projects, clinical practice, as well as basic biological understanding of the largely unexplored changes in cell-cell communication that occurs due to disease and/or associated pharmacological treatment conditions.

**Data Availability Statement:** Code and all raw data are available from GitHub (https://github.com/EffieChantzi/COMBSecretomics.git)

**Funding:** This research work was supported by: - the Swedish Research Council (Vetenskapsrådet, www.vr.se/, grant no. 2017-04655), received by MGG. - the German Research Foundation (Deutsche Forschungsgemeinschaft, www.dfg.de, grant no. 387071423), received by MN. - the European Commission (https://ec.europa.eu/info/index_en, grant no. T1EDK-00120), received by LGA. The funders had no role in study design, data collection and analysis, decision to publish, or preparation of the manuscript.

**Competing interests:** The authors have declared that no competing interests exist.

## Introduction

Living human cells are constantly responding to internal and external stimuli. Every cell in multicellular organisms exchanges messages with itself (autocrine signaling), nearby cells (paracrine signaling), distant cells located in other tissues (endocrine signaling) and the environment in general [1]. For example, when a healthy human cell is stimulated externally by a protein mixture released from other cells, it detects the provocation via an array of surface receptors. Then, after some internal processing, it usually secretes its own protein mixture as a response back. Although such multi-input multi-output (MIMO) chemical communication protocols are of outstanding biomedical relevance and have attracted substantial attention under the name *secretomics* [2–6], there is still very limited knowledge about their modification under different disease and associated treatment conditions. Characterization of disturbances in the MIMO chemical protocols of human cells and infectious agents [7] has therefore the potential to provide fundamental diagnostic and pharmacological insights of key importance for clinical practice, as well as general biological understanding. One inspiring example of what can be achieved in the clinic using this approach can be found in a recent case report; *in vitro* secretomics was employed to find a highly successful personalized drug treatment for a young woman suffering from an unknown form of autoimmune arthritis [8]. As also offered by the novel framework presented in this work, central to this finding was the analysis of secretomic profiles of stimulated (rather than unstimulated) cells, in this case peripheral blood leukocytes obtained from the patient.

Lately, higher-order drug cocktails (i.e., more than two drugs) are increasingly used in the clinic to combat complex and/or co-occurring diseases due to key advantages, such as better efficacy, decreased toxicity and reduced risk of developing resistance [9–17]. Despite the major clinical need for novel and more effective higher-order therapies, the vast majority of drug discovery and development efforts are still limited to either pairwise combination or single-drug treatments. There are outstandingly few reports related to higher-order combination analysis methods and they still come with important shortcomings. Most of them rely on simplistic one-dimensional end point readouts and employ mathematical models that require specific toxicology-rooted mechanistic assumptions about the drug interactions [12–15, 17]. Similarly, COMBImage2 [16], a recently developed live-cell imaging-based computational framework, provides purely phenotypic evaluation of higher-order drug combination effects.

Motivated by this background, we developed COMBSecretomics; a pragmatic methodological framework designed to search exhaustively for second- and higher-order combination treatments that are able to modify, or even reverse malfunctioning secretomic patterns of human cells, when being subject to external naturally occurring and/or disease relevant stimuli (Fig 1). It is based on a standardized reproducible format that could greatly accelerate all studies in this field and also make results obtained by different laboratories directly comparable. This methodological framework could be used together with any experimental platform (typically anti-body based multiplex assays or mass spectrometry) that can provide the required raw data.

We demonstrate the practical use and functionality of COMBSecretomics in terms of an illustrative pharmacological case study focused on cartilage degradation, which is a key feature of osteoarthritis (OA), a highly prevalent chronic disorder and leading cause of disability worldwide [18]. A recently introduced *ex vivo* tissue model of cartilage degradation [19] was employed. 23-dimensional protein release patterns were collected, analyzed and compared after performing an exhaustive combination experiment based on 3 candidate drugs and subsequently stimulating all samples with 3 different naturally occurring protein mixtures.

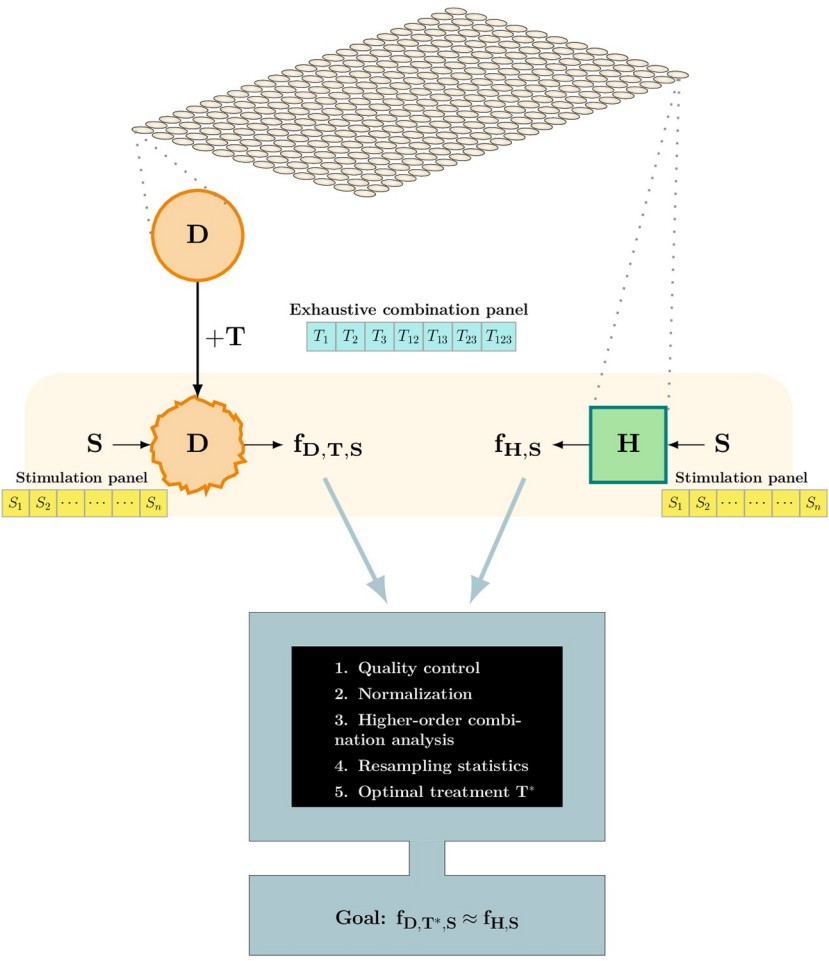

**Fig 1. COMBsecretomics conceptual workflow.** Disease associated (*D*) and healthy (*H*) cells are kept on the same experimental plate to avoid inter-plate variability. *D* cells are exposed to each and every treatment *T* from an exhaustive combination panel; here a panel of all 7 possible treatments using 3 pre-selected drugs $T_1$, $T_2$ and $T_3$ (at fixed concentrations) is shown as an example. *D*-treated and *H* cells are subsequently stimulated with each and every protein mixture *S* from a stimulation panel. Finally, release measurements for a protein panel of interest are collected for both cell types ($\mathbf{f}_{D,T,S}$ and $\mathbf{f}_{H,S}$) using any technology that gives values proportional to the corresponding protein concentrations. The subsequent computational workflow include quality control procedures, normalization of the protein release differences ($\mathbf{f}_{D,T,S} - \mathbf{f}_{H,S}$), model-free higher-order combination analysis and non-parametric resampling statistics. The goal of all these methodological principles is to come up with an optimal (combination) treatment $T^*$ that reverses malfunctioning protein release patterns, meaning $\mathbf{f}_{D,T^*,S} \approx \mathbf{f}_{H,S}$.

## Materials and methods

### *Ex vivo* tissue model

The employed *ex vivo* tissue model aims at providing an experimental platform for OA related cartilage degradation [19]. Cartilage tissue samples were obtained from the femoral heads of a patient (female, 84) undergoing total hip replacement due to fracture with patient's informed consent and all experimental protocols approved by the ethics committee of the KAT General Hospital, Athens, Greece. The femur head was rinsed with PBS, cartilage without subchondral bone was removed and placed into high glucose DMEM (Dulbecco's Modified Eagle Medium) supplemented with 10% FBS, 1% Penicillin/Streptomycin and 1% fungizone (BioCell Technology LLC, Irvine, CA), denoted DMEM*. Cartilage disc samples of 3 mm diameter were created

with a biopsy punch and let to equilibrate in DMEM$^*$ for 24h. In order to obtain healthy ($H$) and disease associated ($D$) samples, the cartilage discs were placed in either fresh DMEM$^*$ or in DMEM$^*$ with collagenase type II, activity 125 units/mg, (MP Biomedicals, Santa Ana, CA) of 2 mg/ml for 24h, respectively. Finally, prior to drug addition, a washing step of 24h in fresh DMEM$^*$ was included. All experiments were conducted in a humidified incubator at 37 ˚C and 5% $CO_2$. In total, 10 $H$ and 73 $D$ samples were created and used (see section "Experimental design" below).

## Stimulations, treatments and multiplex ELISA

In this study, 3 single candidate drug treatments denoted $T_1$, $T_2$, and $T_3$ were used (Table 1), in order to design and perform an exhaustive combination experiment, as defined by Eq (1) below. The treatments were added to the wells (200 $\mu$l DMEM$^*$) and after 24$h$ the supernatant was aspired. Then, fresh DMEM$^*$ (200 $\mu$l) was added to all wells. For a selected subset of wells, stimulations were also added, taken from a set of three alternative protein mixtures, here denoted $S_1$, $S_2$ and $S_3$ (Tables 2 and 3). The individual 7 proteins used in the 3 stimulations were acquired from PeproTech EC Ltd, London. These proteins were selected as they are reported to play driving roles in joint physiology and being involved in OA [19]. The 3 individual drugs and the corresponding concentrations were selected based on a small initial in-house dose response experiment.

24$h$ after stimulation, 80 $\mu$l of the supernatant was retrieved and protein releases were measured with the Luminex xMAP technology offered by FlexMap 3D platform (Luminex Corp. USA). This platform uses an antibody-based suspension array technology to measure the abundance of a predetermined set of proteins [20]. A library of 23 protein releases (PEDF, CXCL11, IL13, ZG16, IL4, GROA, IFNG, CYTC, IL17F, IL12A, TFF3, IL6, ICAM1, IL10, FST, S100A6, CXCL10, PROK1, CCL5, IL20, TNFSF12, MMP9, VEGFA) was measured in the retrieved supernatant.

## Experimental design

In the COMBSecretomics framework, an exhaustive combination experiment covers all plausible subsets among a panel of pre-selected single drug candidate treatments at one fixed concentration each. Following this definition and given $N_t$ single drug candidate treatments, the required number of wells for an exhaustive experiment is expressed as:

$$N_w(N_t) = \sum_{i=1}^{N_t} \binom{N_t}{i} = 2^{N_t} - 1 \tag{1}$$

Thus, if $N_t = 3$ drug treatments are chosen to interfere with 3 different targets related to the disease of interest (like here), an exhaustive experiment exploring all possible ways of modulating these targets requires $N_w(3) = 7$ wells. The experimental design of the pharmacological case study, including 7 different treatments $\{T_1, T_2, T_3, T_{12}, T_{13}, T_{23}, T_{123}\}$ and 3 stimulations $\{S_1, S_2, S_3\}$ (Table 2), is described in Table 3.

**Table 1. Single drugs used as candidate treatments in the pharmacological case study.**

| Annotation | Drug name | Concentration |
|---|---|---|
| $T_1$ | Sorafenib | 10 $\mu M$ |
| $T_2$ | PD169316 | 1 $\mu M$ |
| $T_3$ | Rapamycin | 1 $\mu M$ |

**Table 2. Protein mixtures used as stimulations in the pharmacological case study.**

| Stimulation | Protein mixture |
|---|---|
| $S_1$ | IL1a (50 ng/ml) + TNFa (100 ng/ml) |
| $S_2$ | IL1b (50 ng/ml) + IL8 (100 ng/ml) |
| $S_3$ | BMP2 (100 ng/ml) + TGFb1 (10 ng/ml) + FGF2 (50 ng/ml) |

**Table 3. Exhaustive experimental design.**

| Tissue | Treatment | Stimulation | # Wells | # Replicates |
|---|---|---|---|---|
| D | $\{T_1, T_2, T_3, T_{12}, T_{13}, T_{23}, T_{123}\}$ | $\{S_1, S_2, S_3\}$ | 21 | 2 |
| D | $\{T_1, T_2, T_3, T_{12}, T_{13}, T_{23}, T_{123}\}$ | – | 7 | 3 |
| D | – | $\{S_1, S_2, S_3\}$ | 3 | 2 |
| D | – | – | 4 | – |
| H | – | $\{S_1, S_2, S_3\}$ | 3 | 2 |
| H | – | – | 1 | 4 |
| Blank | – | – | 1 | 4 |

Total number of experimental wells used: $21 \times 2 + 7 \times 3 + 3 \times 2 + 4 + 3 \times 2 + 1 \times 4 + 1 \times 4 = 87$.

## Results

### The COMBSecretomics methodology

**General experimental principle.** COMBSecretomics requires the collected protein release measurements to be obtained as a set of intra-plate replicate experiments performed in any microtiter plate format that allows both healthy (*H*) and disease associated (*D*) cells to be cultivated, treated (*T*) and stimulated (*S*) in parallel to avoid batch variability (Fig 1). Each experimental well results in a *d*-dimensional row vector **f** of raw measurement values, each proportional to the abundance (concentration) of *d* different proteins (Fig 1). The measurements from a whole plate consisting of *N* wells should be stored as a $N \times d$ matrix. In our case study, a $87 \times 23$ matrix was used (S1 Fig in S1 File).

**Quality control.** The collected protein release measurements **f** are pre-processed by COMBSecretomics through a series of tailor made quality control (QC) procedures (see section "Quality Control Explained" in S1 File), in order to eliminate noise and exclude outliers that may trigger misinterpretations. In our case study, the QC procedures resulted in a reduced $76 \times 8$ data matrix (S1–S3 Figs in S1 File). After QC, the median across the intra-plate replicate measurement values **f** is calculated and used to visualize the data (S4 Fig in S1 File).

**Normalization of protein release differences.** COMBSecretomics evaluates systematically secreted protein profiles collected simultaneously for different cell states; disease associated, healthy, treated, untreated, with and without stimulations (Fig 1). The evaluation is based on providing quantitative answers to three pragmatic questions at the level of individual proteins. These three questions, which are thoroughly described in the next section, are designed to determine whether the secretion of a particular protein is affected by the disease (therapeutic need), whether it can be modulated by a treatment (modulation capacity) and whether a treatment can reverse a malfunctioning protein secretion back to the normal/healthy level (restoration capacity).

More specifically, this is done by quantifying differences between the measured sets of protein releases $\mathbf{f}_\alpha$ and $\mathbf{f}_\beta$ of two cell states of interest $\alpha$ and $\beta$, respectively. However, when data is

collected in inter-plate replicates, the protein release differences $\mathbf{f}_\alpha - \mathbf{f}_\beta$ are not directly comparable. Therefore, the collected protein release data should first be normalized per plate. In this way, the results obtained can then be averaged among all plates/batches to reduce experimental variability. The difference in release of protein $k$ between the two cell states $\alpha$ and $\beta$ on plate $p$ is normalized by COMBSecretomics in the following way:

$$r(k, p) = \frac{f_\alpha(k, p) - f_\beta(k, p)}{f_\alpha(k, p) + f_\beta(k, p)} \qquad (2)$$

Here $f_\alpha(k, p)$ and $f_\beta(k, p)$ denote the measurement values for cell states $\alpha$ and $\beta$, respectively. The ratio $r(k, p)$ is restricted to the interval $[-1, 1]$. The value $+1$ is obtained when $f_\alpha(k, p) >> f_\beta(k, p)$, the value 0 when $f_\alpha(k, p) = f_\beta(k, p)$ and the value $-1$ when $f_\alpha(k, p) \ll f_\beta(k, p)$.

**Therapeutic need, modulation capacity and restoration capacity.** As mentioned above, the COMBSecretomics framework is designed to provide quantitative answers to three pragmatic questions related to therapeutic need (Q1), modulation capacity (Q2) and restoration capacity (Q3), at the level of individual secreted proteins. In the following description of the three questions Q1, Q2 and Q3, where Eq (2) is employed, $T_o$ denotes no treatment addition (untreated cell state), while $T_x$ denotes addition of type $x$ treatment ($T_x$-treated cell state). Similarly, $S_o$ denotes the absence of stimulation (unstimulated cell state), while $S_y$ denotes the presence of type $S_y$ stimulation ($S_y$-stimulated cell state).

**Q1) Therapeutic need:** is there any difference in the release of protein $k$ between untreated ($T = T_o$) $D$ and $H$ cells?

To quantify the therapeutic need with respect to protein $k$, the ratio in Eq (2) should be employed with:

a. $\alpha \equiv D, T_o, S_o$ and $\beta \equiv H, T_o, S_o$, when cells are unstimulated ($S = S_o$).

b. $\alpha \equiv D, T_o, S_y$ and $\beta \equiv H, T_o, S_y$, when cells are stimulated ($S = S_y$).
   This ratio determines the release difference of protein $k$ between $D$ and $H$ cells, which essentially defines the therapeutic need with respect to $k$. The bigger this ratio, the bigger the underlying therapeutic need (Fig 2, S5 and S8 Figs in S1 File).

**Q2) Modulation capacity:** is there any treatment ($T = T_x \neq T_o$) that modulates the release of protein $k$ in $D$ cells?

To quantify the modulation capacity of treatment $T_x$, the ratio in Eq (2) should be employed with:

a. $\alpha \equiv D, T_x, S_o$ and $\beta \equiv D, T_o, S_o$, when cells are unstimulated ($S = S_o$).

b. $\alpha \equiv D, T_x, S_y$ and $\beta \equiv D, T_o, S_y$, when cells are stimulated ($S = S_y$).
   A non-zero ratio (either smaller or greater than zero) indicates that treatment $T_x$ seems to modulate the release of protein $k$ in $D$ cells (S6, S9–S11 Figs in S1 File). However, this is not enough in order to understand if the induced modulation by $T_x$ is pointing towards the right direction, meaning to restore the normal release level of protein $k$. This is addressed by quantifying the restoration capacity of treatment $T_x$ defined in question Q3 below.

**Q3) Restoration capacity:** is there any treatment ($T = T_x \neq T_o$) that restores the normal release level of protein $k$ in $D$ cells?

To quantify the restoration capacity of treatment $T_x$, the ratio in Eq (2) should be employed with:

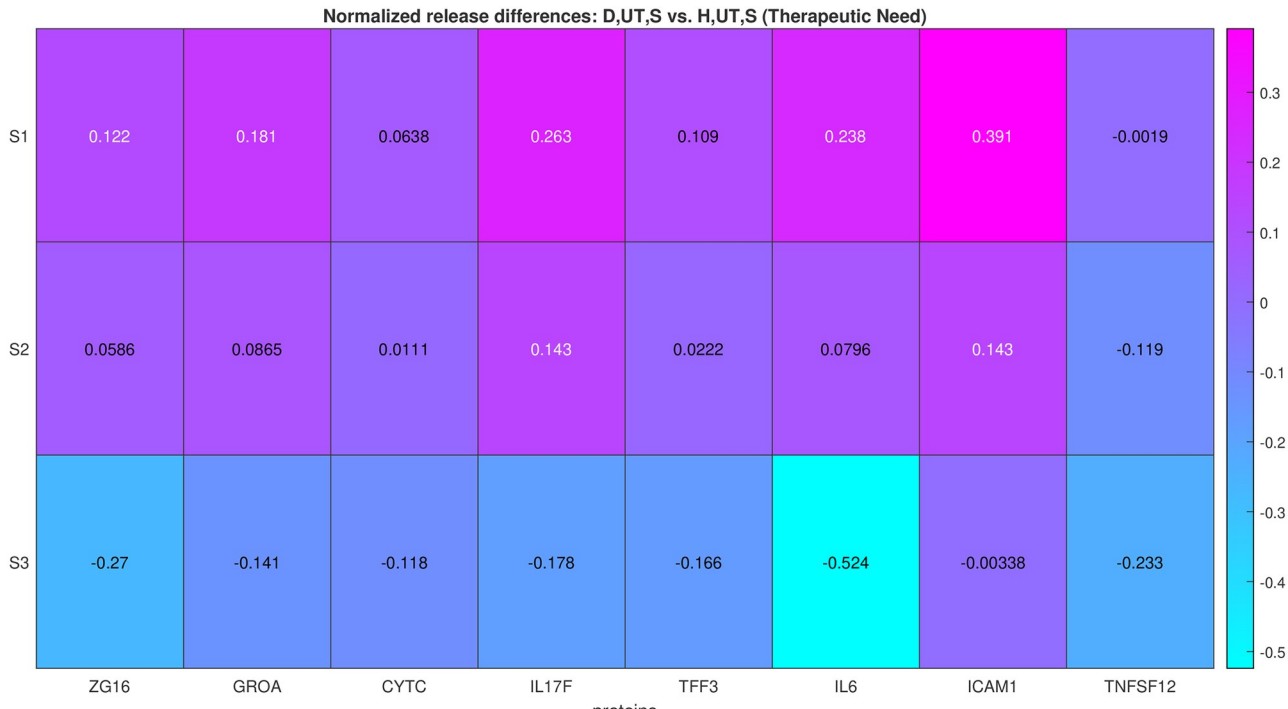

**Fig 2. Therapeutic need quantification.** Example showing the quantitative answers provided to question Q1b, using data from the OA case study presented later in this work. The external stimulations used, denoted $S1$; $S2$; $S3$, are shown in the $y$-axis, while the proteins measured are shown in the $x$-axis. The value in each patch is quantified by employing Eq (2) in the context of answering question Q1b.

 a. $\alpha \equiv D$, $T_x$, $S_o$ and $\beta \equiv H$, $T_o$, $S_o$, when cells are unstimulated ($S = S_o$).

 b. $\alpha \equiv D$, $T_x$, $S_y$ and $\beta \equiv H$, $T_o$, $S_y$, when cells are stimulated ($S = S_y$).
 A zero ratio indicates maximal restoration capacity meaning that treatment $T_x$ is able to restore the normal release level of protein $k$ in $D$ cells, while a non-zero ratio suggests the opposite. In particular, a negative ratio indicates that the release of protein $k$ is higher in $H$ than $D$ cells, while a positive ratio indicates that the release of $k$ is higher in $D$ than $H$ cells (S7, S12–S14 Figs in S1 File).

**Model-free drug combination analysis.** COMBSecretomics offers model-free second- and higher-order combination analysis based on the drug induced restoration capacity of malfunctioning protein secreted profiles (Q3) for exhaustive combination experiments, as defined by Eq (1). Although such brute force experiments may be expensive, they are attractive, as they search exhaustively the combinatorial space. When coupled with proper and robust multivariate data analytics, they can help discriminating higher- from lower-order effects. A higher-order combination treatment should be prioritized when none of its lower-order subsets is able to induce equivalent effects.

**Generalization of the highest single agent principle.** COMBSecretomics performs synergy/antagonism analysis by means of a novel generalized version of the highest single agent (HSA) principle, introduced here for the first time. In general, the HSA concept, known also as Gaddum additivity [21], is well established and mainly used in the context of conventional but relatively simplistic end point cytotoxicity assays. Compared to other widely used neutrality models for synergy/antagonism quantification, such as Bliss independence [22] and Loewe additivity [23], HSA does not require any specific assumption about the drug interactions. On

the contrary, it offers a model-free and straightforward way of identifying useful (i.e., non-antagonistic) combination treatments [24, 25]. This is simply done by checking if the effect induced by the combination is higher than any of the constituent single-drug effects.

In this work, we introduce a generalized version by taking into account not only the single constituents but also all lower-order subsets of a higher-order combination treatment. Let $E(T_x)$ denote the effect of a particular treatment $T_x$. Then, the generalized HSA (GHSA) index, denoted $I_{GHSA}$, for a combination treatment consisting of $N_t$ drugs is defined as:

$$I_{GHSA}(T_{1\cdots N_t}) = \quad min\{E(T_1), E(T_2), \cdots, E(T_{N_t}), E(T_{12}), \cdots,$$
$$E(T_{(N_t-1)N_t}), \cdots, E(T_{2\cdots N_t})\} - E(T_{1\cdots N_t}) \tag{3}$$

Here $I_{GHSA}(T_{1\ldots N_t})$ gives the incremental effect of the combination treatment $T_{1\ldots N_t}$ compared to all single-drug treatments $T_1, T_2, \ldots, T_{N_t}$ and all lower-order combination treatments $T_{12}, \ldots, T_{(N_t-1)N_t}, \ldots, T_{2\ldots N_t}$.

COMBSecretomics determines the total induced effect $E$ of a particular treatment $T_x$ across a panel of $d$ proteins, by means of the restoration capacities obtained from the ratios calculated in question Q3 and collected in the row vector $\mathbf{r}_{Q_3}(T_x)$. For the needs of the generalized GHSA principle, the $d$-dimensional effect of treatment $T_x$ is converted into a scalar in the range [0, 1], by calculating the corresponding normalized $L^1$-norm of the aforementioned $\mathbf{r}_{Q_3}$ vectors as:

$$E(T_x) = ||\mathbf{r}_{Q_3}(T_x)||_1 = \frac{1}{d}\sum_{k=1}^{d}|r_{Q_3}(k, T_x)| \tag{4}$$

$I_{GHSA}$ is restricted to the interval [−1, 1]. The extreme value −1 suggests maximal emergent antagonism, meaning that the higher-order has led to maximally divergent protein release patterns, while one of the lower-order has been able to restore the healthy protein release patterns. The other extreme value + 1 suggests maximal emergent synergy, meaning that the higher-order treatment has been able to restore the healthy protein release patterns, while all of the lower-order treatments result in maximally divergent protein release patterns. The intermediate value 0 indicates that there is no gain by using a higher-order treatment, since the same effect can be achieved by at least one of the lower-order treatments.

**Top-down hierarchical clustering.** COMBSecretomics also employs a data-driven approach to discover prototypical protein release behaviors based on top-down hierarchical clustering using the K-means algorithm at each level [16]. Currently, two hierarchical levels are supported. In this way, the multi dimensional protein release profiles are fully exploited without being compressed into a scalar, like with the GHSA approach described above. The general idea is to split the drug induced protein release patterns into groups with distinct prototypical behaviors, which can be characterized by the user as (un)interesting without any specific assumption about the drug interactions. To render this characterization easier by discerning higher- from lower-order effects (especially for big exhaustive combination panels), a subset search [16] can also be performed upon user request. The goal of this subset search is to narrow down the unique single and/or combination treatments that induce the prototypical protein release profiles for each (sub)group identified. For instance, if one group contains the treatments $\{T_1, T_{23}, T_{13}\}$, then the exhaustive subset search will result only in $\{T_1, T_{23}\}$. Notably, in this case, $T_1$ and $T_{23}$ are ranked equally as they are part of the same group.

The effect of a particular treatment $T_x$ on the release of a specific protein $k$ is determined by its restoration capacity expressed by the ratio calculated in question Q3, here denoted $r_{Q_3}(k, T_x)$. This ratio measures the difference in release of protein $k$ between $T_x$-treated $D$ and

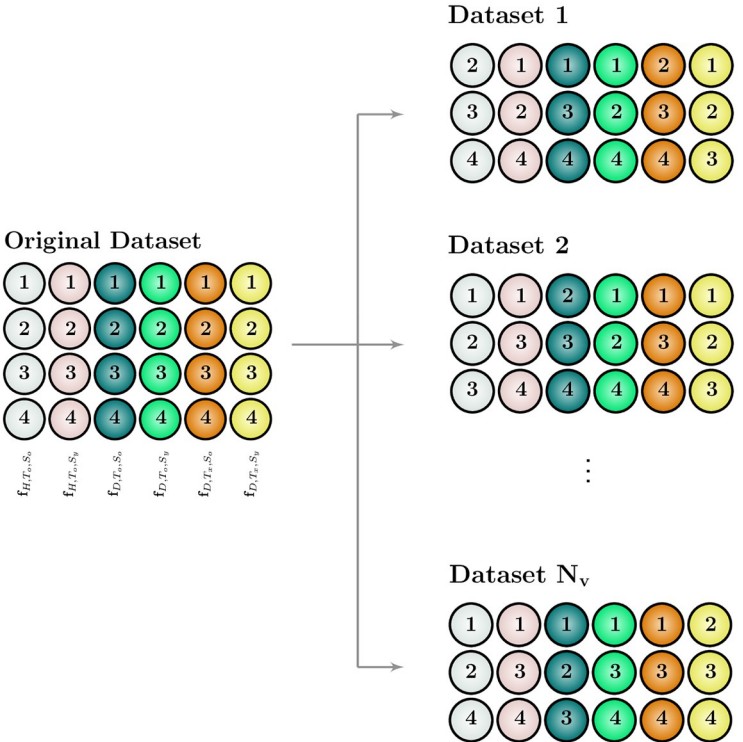

**Fig 3. Miniaturization of the resampling based leave-one-out validation approach per plate.** Left: original collected protein release $d$-dimensional vectors $\mathbf{f}$ for the six different cell states supported by COMBSecretomics $(H, T_o, S_o, \quad H, T_o, S_y, \quad D, T_o, S_o, \quad D, T_o, S_y, \quad D, T_x, S_o, \quad D, T_x, S_y)$. For each cell state, four intra-plate replicate measurements are shown as rows along with the corresponding replicate number. $D$ cells are either untreated $(T_o)$ or treated $(T_x)$ while $H$ cells are untreated $(T_o)$. Both $D$ and $H$ cells are either stimulated $(S_y)$ or not $(S_o)$. Experimental wells of different cell states are colored differently. Right: $N_v$ user-defined validation datasets are automatically created by employing a leave-one-out procedure among the four intra-plate replicate measurements for each cell state.

untreated $(T = T_o)$ $H$ cells. Thus, the total induced effect of $T_x$ across the whole panel of all $d$ proteins is given by the row vector $\mathbf{r}_{Q_3}(T_x)$, which is used for the top-down hierarchical clustering. To ensure that the treatment-induced release patterns are pointing towards the desirable direction, they are jointly visualized with the total therapeutic need, calculated in question Q1 and collected in the row vector $\mathbf{r}_{Q_1}(T_x)$.

**Resampling statistics.** To avoid misinterpretations due to high technical and biological variability, COMBSecretomics employs a non-parametric resampling based validation approach to quantify uncertainty in the combination analysis results obtained. In other words, this statistical procedure adopted by COMBSecretomics helps the user to determine if the combination effects observed could have been obtained without any biological effects. Despite being very valuable, this kind of information is rarely computed and considered. COMBSecretomics provides a simple but powerful validation method, which requires at least two intra-plate replicate experiments per cell state. More specifically, $N_v$ (user-defined) validation datasets are automatically created by randomly leaving out one of the replicate measurements per cell state (Fig 3). Then, the two aforementioned drug combination analysis methods are employed $N_v$ times in total and the different results obtained are compared and used to quantify associated uncertainty measures, described in detail below.

When the GHSA analysis is employed for all different $N_v$ validation datasets (Fig 3), the user gets access to box plots that describe the sampling distribution of the corresponding

combination treatments by means of five summary numbers (Fig 5). In particular, each combination treatment is described by the minimum value, $25^{th}$, $50^{th}$, $75^{th}$ percentiles and maximum value of all different $N_v$ GHSA indices obtained during validation. The closer and more tightly grouped all five aforementioned numbers are, the more certain one should be that the corresponding combination treatment is either synergistic, antagonistic or neutral.

Similarly, when the top-down hierarchical clustering is completed for all different $N_v$ validation datasets (Fig 3), the user gets access to a normalized histogram showing the frequency (%) of all unique clusters/partitions formed at the first hierarchical level (Fig 6a). The fewer different unique partitions with one clearly dominating in frequency over the others, the more certain one should be about the hierarchical partitions obtained. After retrieving the names of drugs/drug combinations of the most dominant clusters/partitions at both hierarchical levels, the corresponding centroids are calculated using all replicates (Fig 6b).

## The COMBSecretomics framework

COMBSecretomics is the result of integrating the aforementioned non-trivial methodologies into a modular framework (Fig 4) that can be systematically employed for secretome-related

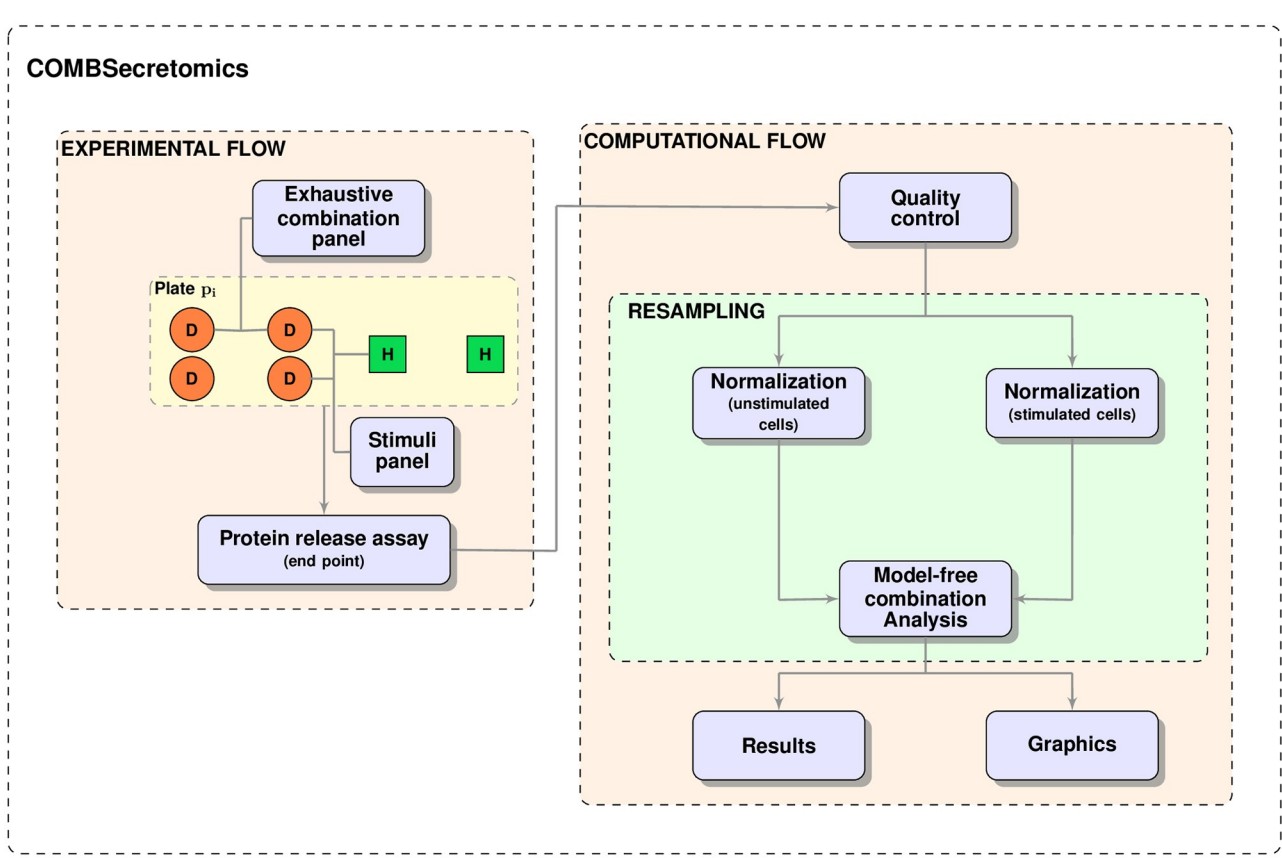

**Fig 4. COMBSecretomics flowchart.** Experimental flow: disease associated (*D*) and healthy (*H*) cells are treated and stimulated in parallel on the same experimental plate $p_i$. An exhaustive combination panel is used for treating *D* cells, while a stimuli panel is employed for both *D* and *H* cells. An end point protein release assay of any sort can be used provided that it gives values proportional to the corresponding protein concentrations. Computational flow: a series of subsequent computational steps are employed for processing the protein release measurement values. Firstly, quality control procedures are employed. Secondly, protein release differences for stimulated and unstimulated cells are normalized per plate $p_i$. Then two model-free combination analysis methods are employed using the normalized protein release differences. Finally, non-parametric resampling statistics are used to quantify uncertainty for the obtained combination analysis results. Graphic and text files with all results are created and saved automatically.

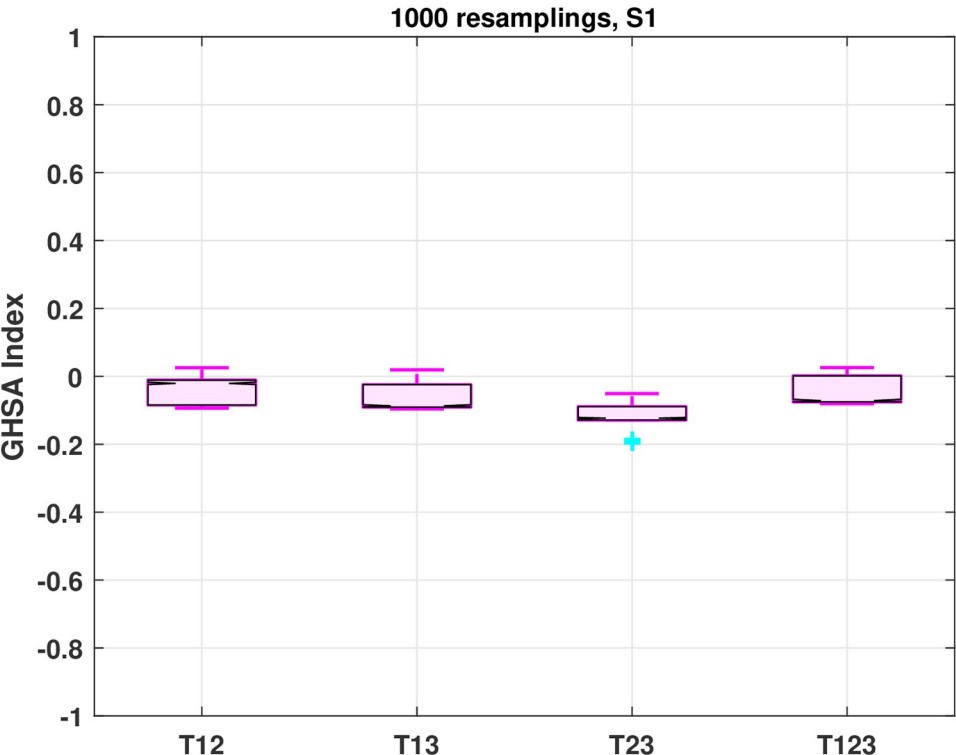

**Fig 5. GHSA analysis for stimulation $S_1$ and $N_v = 10^3$ validation datasets.** All four combination treatments $T_{12}$, $T_{13}$, $T_{23}$, $T_{123}$ are shown on the x-axis. Each combination treatment is represented by a box plot showing the minimum value, $25^{th}$, $50^{th}$, $75^{th}$ percentiles and maximum value of all $N_v$ GHSA indices obtained during a resampling-based leave-one-out validation approach.

second- and higher-order drug combination studies. It is based on a standardized reproducible format, in order to accelerate all studies in this field and facilitate the comparison of results among different laboratories. This methodological framework could be used together with any experimental platform, such as antibody-based multiplex assays or mass spectrometry, which can provide the required raw protein release measurements. Users are only asked to provide the raw data file in CSV file format and a sequence of inputs mainly needed for QC (sections "Example raw data file" and "User-defined inputs" in S1 File).

COMBSecretomics is developed in MATLAB (The MathWorks, Inc., Natick, Massachusetts, United States). It is currently distributed as a MATLAB package and can be deployed with version R2018a or later together with the Statistics and Machine Learning and Bioinformatics toolboxes on Windows, Mac OS X and Linux machines. All modules are well-documented so that future improvements and extensions can be achieved by developers with minimal efforts. For users without access to MATLAB, COMBSecretomics is also provided as a command line tool that can be deployed as a standalone executable on Windows machines. COMBSecretomics is freely available at https://github.com/EffieChantzi/COMBSecretomics.git.

## Pharmacological case study

The practical use and functionality of COMBSecretomics is demonstrated by means of a small pharmacological case study focused on cartilage degradation, a key feature of OA. As described above, 23-dimensional protein release measurements of a recently introduced *ex vivo* tissue model of cartilage degradation [19] were collected and analyzed after performing an

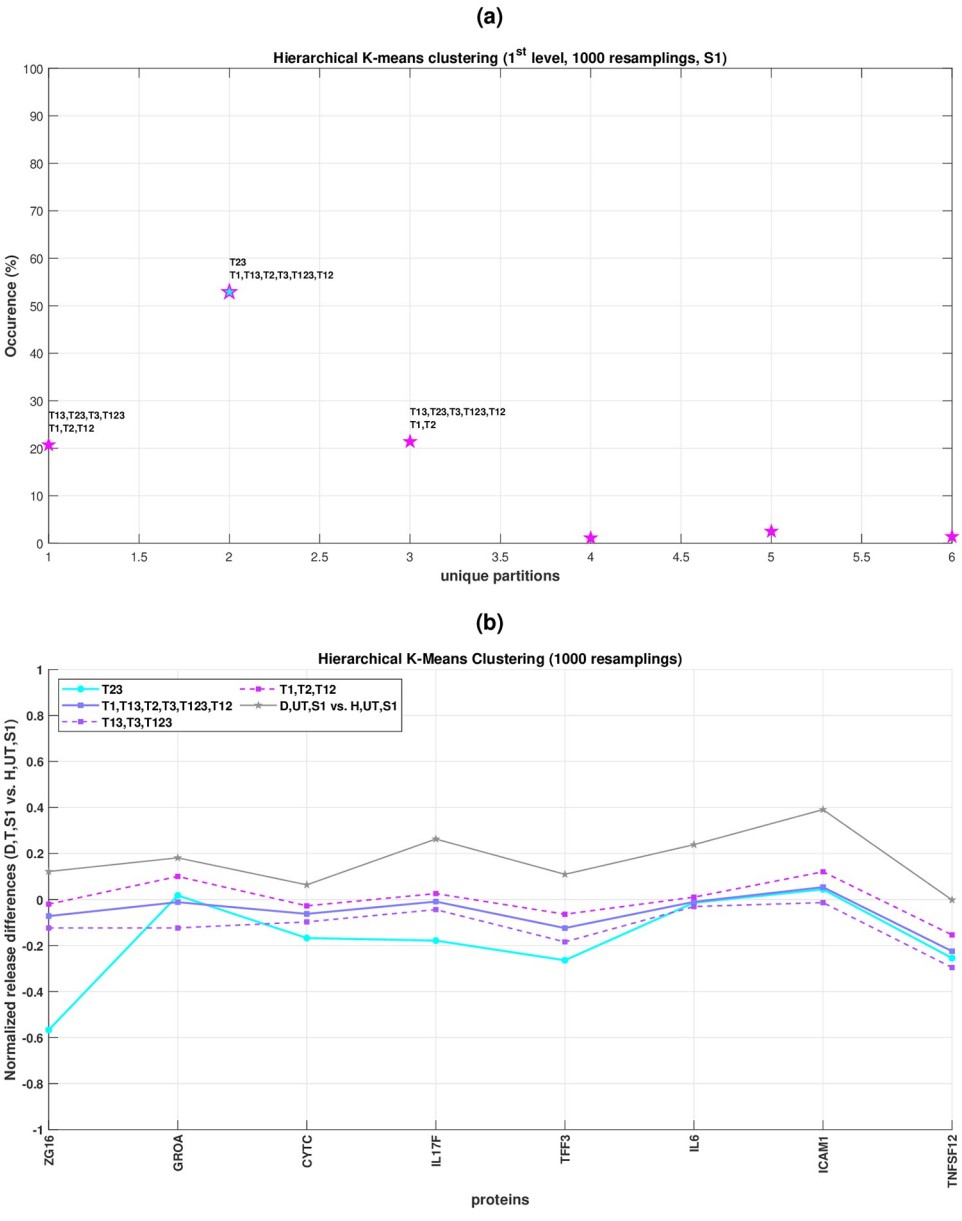

**Fig 6. Top-down hierarchical K-Means clustering for stimulation $S_1$ and $N_v = 10^3$ validation datasets.** (a) Frequency/occurrence (%) for all unique partitions/clusters at the first hierarchical level. Annotations are provided for the three most frequent partitions. (b) Visualization of the clustering results after validation; the most dominant partitions at the first and second hierarchical levels are used. Each line corresponds to the centroid of each (sub-) cluster identified representing its restoration capacity defined by the normalized protein release differences between $D$, treated, stimulated with $S_1$ ($D$, $T$, $S_1$) and $H$, untreated, stimulated with $S_1$ ($H$, $UT$, $S_1$) cells. Solid and dotted lines are used for the first and second hierarchical level respectively. The gray solid line corresponds to the total therapeutic need, meaning the normalized protein release differences between $D$, untreated, stimulated with $S_1$ ($D$, $UT$, $S_1$) and $H$, untreated, stimulated with $S_1$ ($H$, $UT$, $S_1$) cells, which an ideal treatment should eradicate. The legend shows all treatments per (sub-)cluster identified without employing exhaustive subset search since only 7 treatments were used in total.

exhaustive combination experiment based on 3 candidate drugs (Table 1) and subsequently stimulating all samples with 3 different protein mixtures (Table 2). The QC allowed only 8 out of the total 23 protein release measurements to be further used for the drug combination analysis (S1–S3 Figs in S1 File).

Fig 5 shows the combination analysis results with respect to the GHSA method, when both $D$ and $H$ cells were stimulated using the protein mixture $S_1$ (see also S19–S21 Figs in S1 File). The combination treatment $T_{23}$ shows antagonistic behavior, as all five summary values of the corresponding sampling distribution (minimum, $25^{th}$, $50^{th}$, $75^{th}$ percentiles and maximum), obtained during validation, lie on the negative part of the axis. Values very close to zero are mainly observed for the three remaining combination treatments $\{T_{12}, T_{13}, T_{123}\}$, which could be characterized as neutral, suggesting no extra gain by using them.

Fig 6 shows the combination analysis results from the top-down hierarchical clustering approach when both $D$ and $H$ cells were stimulated using the protein mixture $S_1$ (see also S15–S17 Figs in S1 File). The gray line in Fig 6b represents the total therapeutic need, meaning the differences in protein releases between $D$ and $H$ cells that an ideal treatment should eradicate (by bringing them to zero). Two main clusters were identified at the first hierarchical level. The one shown in cyan consists only of $T_{23}$ and seems to induce substantial adverse effects by driving the protein release differences between $D$ and $H$ cells far from zero and specifically, in the opposite direction of the existing total therapeutic need (gray line). Notably, $T_{23}$ appeared to be antagonistic according to the GHSA analysis as well (Fig 5). The second main cluster shown in purple seems promising; its sub-cluster, consisting of treatments $\{T_1, T_2, T_{12}\}$ (lighter purple dotted line), shows great restoration capacity for most of the protein releases. From a combination analysis perspective, the subgroup $\{T_1, T_2, T_{12}\}$ indicates that $T_{12}$ appears to be neutral with respect to the individual treatments $T_1$ and $T_2$ (S18 Fig in S1 File). As shown in Fig 6a, the aforementioned hierarchical partitions were obtained in approximately 55% of all $N_v = 10^3$ validation datasets.

To sum up, both methods indicate that none of the combination treatments $T_{12}$, $T_{13}$, $T_{123}$ seems to offer any substantial benefit compared to the individual treatments $T_1$, $T_2$ (Fig 6b).

## Discussion

COMBSecretomics is the first freely available methodological tool designed to search exhaustively for second- and higher-order combination treatments that can reverse malfunctioning protein release patterns of passive (unstimulated) and active (stimulated) *in vitro* model systems. Its coupling with exhaustive drug combination panels and standardized data analysis methods opens for systematic and reproducible secretome-related drug combination studies. Lately, *in silico* methods gain momentum in drug combination discovery as they are cost efficient and not labor intensive [26, 27]. These methods exploit publicly available omics data, including mainly transcriptomics. Although proteomic datasets are also generated and used in this context, there are currently no publicly available secretomic datasets, where external stimulations have been additionally used to provoke the cells after pharmacological treatment. In this context, COMBSecretomics is unique as it enables the generation and analysis of such complex secretomic datasets, which could in the long term be used by appropriate *in silico* tools. Moreover, the experimental-computational set up of COMBSecretomics could also be employed to confirm *in silico* predictions of promising drug combinations. An approach more similar to COMBSecretomics is adopted by BioMAP; an *in vitro* drug profiling platform based on protein datasets generated from complex (stimulated) primary human cell-based assay systems [28–30]. Although well-established and powerful, BioMAP is commercial and does not provide any methodological and standardized framework for drug combination analyses.

### Things to consider

The composite experimental setup of COMBSecretomics should always be carefully considered. In the particular pharmacological case study presented here, the use of additional/other

stimulations and the use of more/less protein release measurements might have changed the results obtained. Ideally, all secreted proteins should be measured, in order to get a comprehensive picture of what changes the tested drugs and drug combinations are able to achieve. Furthermore, a larger panel of drugs would also have resulted in different/extended results, covering much more drug combinations than tested and reported in this work. However, the actual raw protein release profiles obtained for the current single drugs and drug combinations would not have changed (except for experimental variability). In other words, repeating the same experiment and analyzing the corresponding generated data should give very similar results as the ones presented here. Moreover, the current findings cannot be used to disentangle if the observed effects are exclusively due to modulations induced by the single drugs/drug combinations or they also involve interactions between the stimulating proteins and the added drugs. The complexity of the tissue/cell model system employed is also expected to affect the results obtained. Moreover, there is no information provided about potential drug-drug interactions, since the COMBSecretomics framework is employed in the context of *in vitro* experiments based on a carefully selected set of drug candidates. Thus, such issues are beyond the scope of the novel framework COMBSecretomics introduced here.

## Facts and limitations

Based on a highly standardized experimental-computational procedure, COMBSecretomics helps to simultaneously pinpoint important causal biological and pharmacological combination effects at the level of individual proteins and protein profiles secreted. Notably, thanks to a resampling based statistical procedure, the risk of interpreting ordinary technical and biological variability as a real biological effect is reduced. The framework is not designed to provide explicit information or hypotheses regarding causality and mechanistic effects. It is rather designed to systematically evaluate secreted protein profiles collected simultaneously for different cell states; disease associated, healthy, treated, untreated, with and without stimulations. This evaluation is based on providing quantitative answers to three pragmatic questions at the level of individuals secreted proteins.

In particular, these questions are designed to determine whether the secretion of a particular protein is affected by the disease (therapeutic need), whether it can be modulated by a treatment (modulation capacity) and whether a treatment can reverse a malfunctioning protein secretion back to the normal/healthy level (restoration capacity). Then, COMBSecretomics performs model-free second- and higher-order combination analysis based on the drug induced restoration capacity. This is firstly achieved by introducing a tailor made generalization of the HSA principle, which quantifies the additional biological/pharmacological effect achieved at the global protein secretion level, when a set of drugs is used in combination. Secondly, this systemic analysis is also complemented with a data mining approach based on top-down hierarchical clustering, as a means to identify prototypical drug induced protein secretion patterns and discern higher- from lower- and single-drug effects.

COMBSecretomics quantifies how much the measured protein release responses to stimulations are changing between before and after drug exposure, thereby offering causal information of a kind that almost never is neither obtained nor reported. Thus, the causal information provided is reflecting to which extent a drug combination can transform the disease model, so that it responds differently to the same stimulations after the pharmacological treatment. Ideally, the pharmacologically induced responses should be similar to the healthy cellular responses. These results provide concrete valuable information. For example, high restoration capacity indicates a promising treatment, whereas low restoration capacity demonstrates a

treatment that should be avoided in the clinic (as it drives the protein release to diverge even more from the normal levels). Mechanistic interpretations of the results in the context of associated drug targets and involved biochemical pathways/processes is beyond what COMBSecretomics is designed to address. Therefore, for more detailed biological, pharmacological and mechanistic effects, a post-analysis including additional wet lab experiments focused on the most outstanding results, is required.

## Conclusion

In brief, COMBSecretomics

- fills in an important lack by enabling systematic test tube analysis of how second- and higher-order drug combinations can affect secretomic patterns of human cells (meaning protein concentration profiles secreted by primary patient cells as well as patient derived cell lines), when being subject to natural and/or disease relevant protein stimulations. In contrast to state-of-the-art analysis methods, which are limited to drug pairs and/or unstimulated cells, this framework for the first time enable extensions along both these fronts.

- is the first methodological framework developed to search exhaustively for second- and higher-order drug combination treatments that can modify, or even reverse malfunctioning secretomic patterns of human cells. In particular, this generic framework could be used for individualized drug combination therapy selection in the clinic, drug discovery and development projects, as well as understanding the largely unexplored changes in cell-cell communication that occur due to disease and/or pharmacological treatments.

- reflects the need and feasibility for standardization of the necessary but non-trivial integrated mix of experiments and analysis required in the context of secretomics and higher-order drug combination analysis. Only through this kind of frameworks, proper comparisons of results across different laboratories will be possible.

- comes with two novel model-free methods for drug combination analysis of the secretomic patterns collected. The first is a generalization of the highest single agent method, which is designed for drug pairs and cytotoxicity, to higher-order combinations and multi dimensional readouts. The second model-free method is based on top-down hierarchical clustering of the secretomic profiles collected, which returns a hierarchy of prototypical secretion patterns and the corresponding most similar real patterns collected.

- provides systematic quality controls at multiple levels to eliminate outliers and non-parametric resampling statistics to quantify uncertainty in the results obtained.

Taken together, COMBSecretomics consists, to the best of our knowledge, the first pragmatic framework that enables secretome-related second- and higher-order drug combination analyses. To maximize the potential of COMBSecretomics for basic biological understanding and guide systematic combination studies for disease areas where there are still unmet diagnostic and therapeutic needs, we envision further development and refinements. These may include support for stimuli-related exhaustive experimental designs, replacement of the currently employed steady cell states (24h of stimulation) with time series measurements and development of a laboratory information management system. As a natural extension, this generic and modularized tool could also easily be adjusted for chemical biology and toxicology research, where characterization of chemical mixtures is a highly important but largely neglected area.

## Supporting information

**S1 File. Quality control procedures including S1–S21 Figs.**
(PDF)

## Author Contributions

**Conceptualization:** Efthymia Chantzi, Mats G. Gustafsson.

**Data curation:** Efthymia Chantzi, Michael Neidlin.

**Formal analysis:** Efthymia Chantzi, Mats G. Gustafsson.

**Funding acquisition:** Michael Neidlin, Mats G. Gustafsson.

**Investigation:** Efthymia Chantzi, Michael Neidlin, Leonidas G. Alexopoulos, Mats G. Gustafsson.

**Methodology:** Efthymia Chantzi, Michael Neidlin, Leonidas G. Alexopoulos, Mats G. Gustafsson.

**Project administration:** Michael Neidlin, Mats G. Gustafsson.

**Resources:** Michael Neidlin, George A. Macheras, Leonidas G. Alexopoulos, Mats G. Gustafsson.

**Software:** Efthymia Chantzi.

**Supervision:** Mats G. Gustafsson.

**Validation:** Efthymia Chantzi, Mats G. Gustafsson.

**Visualization:** Efthymia Chantzi.

**Writing – original draft:** Efthymia Chantzi, Mats G. Gustafsson.

**Writing – review & editing:** Efthymia Chantzi, Michael Neidlin, George A. Macheras, Leonidas G. Alexopoulos, Mats G. Gustafsson.

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
