## [Decision Letter · Decision Letter 0]

18 Mar 2020

PONE-D-20-06015

COMBSecretomics: a pragmatic methodological framework for higher-order drug combination analysis using secretomics

PLOS ONE

Dear Dr. Chantzi,

Thank you for submitting your manuscript to PLOS ONE. After careful consideration, we feel that it has merit but does not fully meet PLOS ONE’s publication criteria as it currently stands. Therefore, we invite you to submit a revised version of the manuscript that addresses the points raised during the review process.

We would appreciate receiving your revised manuscript by May 02 2020 11:59PM. To enhance the reproducibility of your results, we recommend that if applicable you deposit your laboratory protocols in protocols.io, where a protocol can be assigned its own identifier (DOI) such that it can be cited independently in the future. For instructions see: http://journals.plos.org/plosone/s/submission-guidelines#loc-laboratory-protocols

We look forward to receiving your revised manuscript.

Kind regards,

Le Zhang

Journal Requirements:

Please ensure that your manuscript meets PLOS ONE's style requirements, including those for file naming. The PLOS ONE style templates can be found at http://www.plosone.org/attachments/PLOSOne_formatting_sample_main_body.pdf and http://www.plosone.org/attachments/PLOSOne_formatting_sample_title_authors_affiliations.pdf

Reviewers' comments:

Reviewer's Responses to Questions

**Comments to the Author**

1. Is the manuscript technically sound, and do the data support the conclusions?

Reviewer #1: Yes

Reviewer #2: Yes

2. Has the statistical analysis been performed appropriately and rigorously? 

Reviewer #1: Yes

Reviewer #2: N/A

3. Have the authors made all data underlying the findings in their manuscript fully available?

Reviewer #1: Yes

Reviewer #2: Yes

4. Is the manuscript presented in an intelligible fashion and written in standard English?

Reviewer #1: Yes

Reviewer #2: Yes

5. Review Comments to the Author

Reviewer #1: This paper reports a pragmatic methodological framework for higher-order drug combination analysis (COMBSecretomics) aiming on characterization of disturbances in the multi-input multi-output (MIMO) chemical protocols of human cells and infectious agents. The conceptual workflow, experimental details, novel model-free methods for drug combination analysis of the secretomic patterns collected have been carefully elaborated and the total framework was validated by means of a proof-of-principle pharmacological study related to cartilage degradation. This paper is novel for analyzing the modulation of complex systems by quantitative means. Therefore, I suggest that this manuscript should be reconsidered after a minor revision.

1. In the manuscript, the pharmacodynamic effects of drug combinations were evaluated by a subtle experimental framework including tissue model, candidate drugs, stimulating proteins, releasing proteins. In the pharmacological study related to cartilage degradation, the cartilage disc samples, 3 candidate drugs, 7 stimulating proteins and 23 releasing proteins were applied. All these factors could have a significant effect on the final results. It’s suggested that the standard for selection of these factors should be furthered discussed in the Discussion part.

2. Considering the complexity of tissue or cell systems, the drug and the stimulating proteins may both affect the expression of the releasing proteins. Therefore, it’s suggested that the solo or combination effect of these two factors on releasing proteins should be discussed in the Discussion part.

3. The information of some literatures in References section was incomplete. For example : 24. Vlot AHC, Aniceto N, Menden MP, Ulrich-Merzenich G, Bender A. Applying synergy metrics to combination screening data: agreements, disagreements and pitfalls. Drug Discov Today. 2019;.

Reviewer #2: Chantzi et al reported a novel framework designed to exhaustively search for second- and higher-order mixtures of candidate treatments that can modify, or even reverse malfunctioning secretomic patterns of human cells. It includes a tailor-made generalization of the highest single agent principle and a data mining approach based on top-down hierarchical clustering. It is used as a proof-of-principle pharmacological study related to cartilage degradation. The concept of this work is novel and the results are interesting. However there are some concerns that should be addressed.

1. The higher-order drug combination analysis is based on data processing of secretomics. It misses the causality and mechanism of the combination effects of drug. So it is difficult to identify the drugs which is just statistically correlated with each other or not.

2. The Pharmacological case study is written in a way of data processing but not a prototype exhibition of the value in pharmacological study. How does the result connect with the real biological effects? Does the result make sense?

3. The current work lacks systematic evaluation. Methods for pharmacological researches need to be validated statistically using some benchmarks.

4. The Normalization of protein release differences is a critical step of the method. I miss some explanation of the effectiveness in eliminating batch effects.

5. The current writing is not friendly to the users of the method. It can provide more biological insights if the work can exhibits the relationship between the statistical results and real biological effects.

6. PLOS authors have the option to publish the peer review history of their article (what does this mean?). If published, this will include your full peer review and any attached files.

Reviewer #1: No

Reviewer #2: No

---

## [Author Response · Author response to Decision Letter 0]

23 Apr 2020

Reviewer #1

1. Answer: We agree that the use of additional/other stimulations and the use of smaller/bigger set of protein measurements might have changed the results. In addition, using a larger panel of drugs would also have resulted in different/extended results, covering a larger set of drug combinations than reported here. However, the actual raw data profiles obtained for the current single drugs and drug combinations tested here would of course not have changed (except for experimental variability). In other words, a separate analysis using only them should give very similar results as the ones presented here. All these matters have now been elaborated on in the “Discussion” in the newly added section “Things to consider”. 

2. Answer: We agree that there might be interactions caused by the drugs used for treatment and the stimulating proteins, as an alternative to the assumed situation, where the drugs modulate the cellular response to the stimulating proteins presented after drug treatment. As an extension to question 1 above, this matter has now been elaborated on in the “Discussion” in the newly added section “Things to consider”.

3. Answer: We are sorry for this mistake and we are thankful to the reviewer for pointing this out. This error has now been corrected.

Reviewer #2

1. Answer: First of all, we would like to stress and clarify that COMBSecretomics is not at all about finding statistical correlations, The statistical (resampling) part of this framework is only there to reduce the risk of interpreting ordinary technical and biological variability as a real biological effect. More specifically, we are quantifying the uncertainty in the results obtained, in order to exclude the possibility that the results could have been obtained even without any biological effects. 

 Secondly, we would like to stress and clarify that COMBSecretomics actually provides very detailed information about biological/pharmacological effects at the level of individual proteins secreted. We agree that the framework is not designed to provide explicit information or hypotheses regarding causality and mechanistic effects but rather designed to provide quantitative answers to three quite pragmatic questions related to “Therapeutic need” (Q1), “Modulation capacity” (Q2), and Restoration capacity” (Q3), at the level of individual proteins released. At the same time the framework also provides quantitative answers at the system level in terms of full protein release profiles, where the extended HSA (Highest Single Agent) principle is employed to quantify the additional biological/pharmacological effect provided when a set of single drugs are used in combination. In addition, this systemic analysis is complemented with top-down hierarchical clustering of the changes in protein release profiles as a means to rapidly and easily identify archetypical systemic biological/pharmacological effects and relate them to the HSA analysis results. A final important fact in this context is that the framework is focused on how protein release responses to stimulations are changing between before and after drug exposure, thereby offering causal information of a kind that almost never is neither obtained nor reported. Taken together, the framework actually provides quite detailed answers that indeed may be very valuable in terms of formulating hypotheses about plausible causalities and mechanisms. However, this kind of post-analyses is challenging and typically requires follow up experiments that are beyond the scope of the framework and the simple proof-of-concept example. 

 Thirdly, if the reviewer´s question also concerns drug-drug interaction effects in this context, the answer is that pinpointing drug-drug interactions is beyond the scope of the current work. COMBSecretomics will indeed sift out antagonistic effects but the reasons when finding such an effect is a topic for post-analyses and follow up experiments (unless there is already a well known drug-drug interaction reported for a particular antagonistic result obtained). In the particular proof-of-principle case study presented in the paper, the main antagonistic pair identified contains one experimental drug and one clinically used drug. Therefore, there is no guidance from public drug-drug interaction databases and scientific literature regarding if the observed antagonistic effect should be expected by some form of drug-drug interaction or not.

 Finally, here we would also like to stress that the quite unique analyses provided by COMBSecretomics, where protein release patterns before and after pharmacological exposure are compared, provides a type of causal information almost never reported: For each combination, quantitative causal information is provided reflecting to which extent a drug combination can transform the disease model, so that it responds differently to the same natural protein stimulations after the pharmacological treatment. However, we are uncertain what exactly the reviewer means by causality in this context .

 All these matters have now been addressed in the “Discussion” in the newly added section “Facts and limitations”, in order to reduce potential confusion to the reader regarding what the framework can be used for.

2. Answer: We are sorry that the reviewer thinks we have not presented a prototype exhibition of the value of the pharmacological case study, but rather done it in a “data processing” way. We believe that the current way of presenting it actually pinpoints the main pharmacological value directly, expressed in terms of answers to the three pragmatic questions Q1, Q2 and Q3 posed in the manuscript provided. As already discussed in our answer to question 1 above and elsewhere, we fully agree that there might be much to gain from a more detailed analyses of the most outstanding combinations found in terms of antagonistic effects and synergistic effects, if they actually exist in terms of Q1, Q2 and Q3. However, this kind of added values will be the result of a post-analysis that typically also will include additional validation and follow-up wet lab experiments to disentangle different hypotheses that will arise. Therefore, this kind of analyses goes far beyond the framework introduced and the small proof-of-principle study provided.

 Thus, as already explained in our answers to question 1 above, we are indeed quantifying and reporting real biological effects, both at the level of individual proteins secreted and the systemic level in terms of changes in protein release profiles. Therefore, the results provided actually make a lot of sense. For example, if we find a combination that has high “restoration capacity” it would mean a promising treatment candidate, whereas finding a combination with an antagonistic “restoration capacity” would indicate a treatment that definitely should be avoided in the clinic (as it drives the protein release to diverge even more from the normal levels). To which extent this makes sense from a mechanistic point of view in terms of current understanding of associated drug targets and involved biochemical pathways/processes is far beyond the current scope of the framework COMBSecretomics.

 All these matters have now been addressed in the “Discussion” in the newly added section “Facts and limitations”, with the aim to make the practical potential and limitations of COMBSecretomics more obvious to the reader. Moreover, we have added a new paragraph and a new figure in section “Normalization of protein release differences” of the “Results”, as an attempt to present the methodology of COMBSecretomics in a more user-friendly way, as the reviewer wishes.

3. Answer: As this is the first framework of its kind publicly available, and there are no known successful drug combination treatments for osteoarthritis reported in the literature suitable for validation/benchmarking, there is simply nothing to compare with. If we have overlooked something along these lines, we would greatly appreciate to get a pointer/link to appropriate data sets or methods to compare with. This matter is also mentioned in the “Discussion”and “Conclusion”.

4. Answer: We fully agree that this is a crucial step and this is exactly why we use this particular normalization approach, where the normalization is done for each plate separately in order to eliminate batch (plate-to-plate) variability. Its effectiveness is because the normalization is performed per plate, in order for the resulting data to be directly comparable across all plates/batches. In the manuscript text, we have now provided additional details and stressed more clearly the importance of this normalization. More specifically, we have rephrased the corresponding text in section “Normalization of protein release differences”. 

5. Answer: We are sorry that the reviewer perceives the current description of the framework and the results provided much less user-friendly than they actually are in practice. We believe that this is only related to the fact that we introduce a novel standardized framework (including a novel terminology and novel methodology like the higher-order analysis) designed to mainly give answers about real biological effects related to the three quite pragmatic questions Q1, Q2, and Q3 posed in the main text. These questions are mainly intended to detect the most outstanding biological/pharmacological effects. The associated statistical analyses (resampling) is only there to avoid misinterpretations caused by ordinary experimental and biological variability.

 In other words, as we have stated multiple times in some of our previous answers above to the reviewer, COMBSecretomics does not provide statistical results but actually real biological effects in terms of changed secretomic patterns due to single drug and combination exposures. In order to narrow down the results to practically meaningful real biological effects we have introduced and defined the three questions related to “Therapeutic need” (Q1), “Modulation capacity” (Q2), and Restoration capacity” (Q3). Although this terminology and these questions perhaps do not align exactly with conventional pharmacological and biological terminology/thinking, we strongly believe that these are practically very useful and understandable quantities to report as the first results from this kind of studies. For more detailed biological/pharmacological effects at the levels of hypotheses regarding causality and mechanisms, a post-analysis will be required that will be focused on the most outstanding biological/pharmacological effects observed in terms of answers to the questions Q1, Q2 and Q3. 

 All these matters have now been addressed in the “Discussion” in the newly added section “Facts and limitations”, with the aim to make the practical potential and limitations of COMBSecretomics more obvious to the reader. Moreover, we have also clarified the goal of the statistical (resampling) part by rephrasing the corresponding text in section “Resampling statistics” of the “Results”. These changes in combination with other additions and modifications related to the reviewer’s concerns above are aiming at presenting the methodology of COMBSecretomics in a more user-friendly way, as the reviewer wishes.

---

## [Editor Report · Decision Letter 1]

27 Apr 2020

COMBSecretomics: a pragmatic methodological framework for higher-order drug combination analysis using secretomics

PONE-D-20-06015R1

Dear Dr. Chantzi,

We are pleased to inform you that your manuscript has been judged scientifically suitable for publication and will be formally accepted for publication once it complies with all outstanding technical requirements.

With kind regards,

Le Zhang

Academic Editor

PLOS ONE
---

## [Editor Report · Acceptance letter]

29 Apr 2020

PONE-D-20-06015R1 

COMBSecretomics: a pragmatic methodological framework for higher-order drug combination analysis using secretomics 

Dear Dr. Chantzi:

I am pleased to inform you that your manuscript has been deemed suitable for publication in PLOS ONE. Congratulations! Your manuscript is now with our production department. 

With kind regards,

on behalf of

Dr. Le Zhang 

Academic Editor

PLOS ONE